

# Transcriptome analysis of substrate temperature effects on adventitious root formation in peach rootstocks

Fan Zhang, Hong Wang, Chenbing Wang, Xiaoshan Wang, Jiaxuan Ren and Meimiao Guo

Institute of Forestry, Fruits and Floriculture, Gansu Academy of Agricultural Sciences, Lanzhou, China

## ABSTRACT

The propagation of peach rootstocks, particularly adventitious root (AR) formation, is influenced by multiple factors, with substrate temperature being crucial. This experiment studied the differential gene expression patterns of GF677 rootstock cuttings treated with 200 mg L$^{-1}$ indole-3-butyric acid (IBA) under various substrate temperatures (ambient temperature (CK), 19 °C, 22 °C, 25 °C, and 28 °C) and cutting periods (7, 14, and 21 days). The results showed a maximum rooting rate of 91% when assessed at 40 days under 25 °C, while RNA sequencing was performed at earlier stages (7, 14, and 21 days). The highest number of differentially expressed genes (DEGs) observed between 22–25 °C. Therefore, the optimal substrate temperature for propagation was determined to be 25 °C. Gene ontology (GO) and Kyoto Encyclopedia of Genes and Genomes (KEGG) analysis highlighted "starch and sucrose metabolism (photosynthesis processes)" and "plant hormone signal transduction (especially auxin)" as enriched pathways. Specifically, 26 plant genes (*ARFs*, *LBDs*, *SAURs*, and *GH3*) and 22 AR formation-related genes (*AUR3*, *LRP1*, *RGF1*, *AIR9*, *AP2*, and *NAC*) were identified from these DEGs. Weighted gene co-expression network analysis (WGCNA) clarified the involvement of various transcription factors (*WRKYs*, *ERFs*, *NACs*, *bHLHs*, *bZIPs*, and *MYBs*) in AR formation. These findings indicate significant differences in gene expression under different combinations of substrate temperatures and cutting periods. Overall, this study enhances our understanding of the molecular mechanisms underlying peach rootstocks asexual reproduction.

# INTRODUCTION

Seedling rootstocks are predominantly used in peach tree propagation due to their low-cost seeds and easier sexual reproduction compared to cutting propagation (*Sabbadini et al., 2015*; *Adaskaveg, Schnabel & Förster, 2008*). However, genetic heterogeneity from seed propagation can lead to the loss of crucial traits, thereby potentially affecting orchard productivity (*Lavania, Srivastava & Lavania, 2010*). In addition, compared to self-rooted plants, grafted plants have multiple advantages, including enhanced resistance to biotic and abiotic stress (*Melnyk, 2016*), improved field performance (increased yield, earlier maturity), and optimized orchard horticultural management practices (*Ming et al., 2024*).

Corresponding author
Fan Zhang, zhfan528@163.com

Nonetheless, certain tree species face challenges in forming adventitious root (AR) during cutting propagation (*Druege et al., 2019*). ARs play a crucial role in plant growth and development by facilitating water and nutrient absorption, enhancing plant stability, and increasing stress resistance (*Druege, Franken & Hajirezaei, 2016*; *Ahkami, 2023*). However, peach trees are among the tree species that exhibit difficulty in rooting through cutting propagation, resulting in a consistently low rooting rate throughout the propagation process. Additionally, the rooting of peach branch cuttings is influenced by numerous regulatory factors (*Beckman, Nyczepir & Myers, 2006*), with temperature being a significant determinant. Maintaining an optimal substrate temperature conducive to rooting is crucial during the cutting process (*Tsipouridis, Thomidis & Michailides, 2005*). Therefore, it is particularly important to deeply explore and determine the appropriate substrate temperature for adventitious root formation in the practice of cutting propagation.

GF677 (*P. amygdalus* × *P. persica*) is a peach rootstock variety developed in France during the 1960s. This variety has a well-developed root system, robust growth, and resistance to calcium alkaline soil iron deficiency chlorosis, replant disease, and drought. Furthermore, its excellent genetic traits must be maintained through vegetative propagation (*Tsipouridis, Thomidis & Michailides, 2005*; *Tsipouridis, Thomidis & Bladenopoulou, 2006*; *Ricci et al., 2023*). However, there is no established vegetative propagation production technology system for GF677 rootstocks in China, and the hardwood cuttings have a low rooting rate, which restricts the development of the peach industry. Therefore, there is an urgent need to develop a vegetative propagation technology system for GF677 rootstocks to solve the problem of difficult peach grafting.

Auxin regulates AR formation through complex molecular mechanisms involving transport, reception, and signaling pathways (*Veloccia et al., 2016*). Key auxin-responsive gene families include auxin/indole-3-acetic acid (AUX/IAA) (repressors of ARFs), Gretchen Hagen3 (GH3) (maintains auxin homeostasis), and Small Auxin Up RNA (SAUR) (regulates cell expansion and root development) (*Druege, Franken & Hajirezaei, 2016*; *Wang et al., 2024a*; *Wang et al., 2024b*). ARFs activate auxin response elements to promote root initiation (*Wilmoth et al., 2005*), while LBD proteins act downstream of ARFs in AR formation (*Lee et al., 2019*; *Zhu et al., 2016*). In *Arabidopsis thaliana*, genes like *LRP1*, *RGF*, and *AIR12* link auxin signaling to root development (*Singh et al., 2020*; *Shinohara, 2021*). Additionally, transcription factors (*e.g.*, *AP2/ERF*, *bHLH*, *WRKY*, *NAC*, *MYB*, *bZIP*) mediate early transcriptional regulation of AR (*Ai et al., 2023*).

In this study, we selected the peach GF677 rootstocks as the subject of research to deeply investigate the effects of different cutting periods, and substrate temperature conditions on the process of adventitious root formation. Subsequently, we employed transcriptome sequencing technology and weighted gene co-expression network analysis (WGCNA) to analyze differentially expressed genes (DEGs) that emerged under different substrate temperatures (ambient temperature (CK), 19 °C, 22 °C, 25 °C, and 28 °C) and cutting periods (7, 14, and 21 days). We identify optimal substrate temperature conditions and key regulatory genes underlying AR development. This research provides a valuable theoretical reference for this field.

## MATERIALS & METHODS

### Plant material and sample preparation

This experiment was conducted under controlled conditions in a greenhouse at the Gansu Academy of Agricultural Sciences to investigate the hardwood cutting propagation of the peach GF677 rootstocks. The greenhouse temperature is 18/5 °C (day/night), the photoperiod is nine hours of light/15 h of darkness, and the humidity is 55–75%. We will take the rootstock branches in the middle of November 2024. The cuttings were selected from the middle and upper portions of current-year branches, ensuring they were free of pests and diseases and had diameters ranging from 0.5 to 1.0 cm, and after being stored in sand for one month, hardwood cuttings will be taken for propagation. The optimal length of the cuttings was 15 to 20 cm. A cutter was used to create a 40 to 45° bevel at the base near the basal bud. The top of the cutting was cut flat, and it was advisable to apply a plant wound healing agent to the flat surface. The substrate consisted of a 1:1:1 volume ratio of perlite, peat moss, and vermiculite. The cuttings were dipped in a solution of 200 mg $L^{-1}$ indole-3-butyric acid (IBA) for 20 s before planting, while the control group was dipped in water (*El-Boray et al., 1995*; *Eliwa & Wahba, 2018*; *Zhang & Wang, 2018*; *Zhang et al., 2013*). They were heated using electric heating elements (ambient temperature (CK), 19 °C, 22 °C, 25 °C, and 28 °C) and the rooting rates of the five treatments were counted after 40 days. Sampling was conducted on the 7th day (labeled as CT1, CT2, CT3, CT4, and CT5), the 14th day (labeled as, CT6, CT7, CT8, CT9 and CT10), and the 21st day (labeled as, CT11, CT12, CT13, CT13 and CT14) for RNA sequencing. CT0 indicates that the cuttings were treated with IBA and were planted after 0 days at ambient temperature. CT1, CT6 and CT11 represent cuttings treated with IBA and planted after 7, 14 and 21 days respectively at ambient temperature. For detailed information on the experimental design of this study, please refer to Table 1. In this study, three biological replicates were set for each treatment to ensure the reliability of the results. Each replicate contained 30 randomly selected healthy cuttings, which were taken from multiple plants in the greenhouse to reduce genetic variability.

### RNA library construction and high-throughput sequencing

We sampled the phloem from two cm of the base of each cutting, mixed them evenly and extracted RNA. Following the manufacturer's instructions, total RNA was extracted from the plants using a Pure Plant RNA Extraction Kit (Tiangen, China). The concentration and purity of the extracted RNA were measured using a NanoDrop 2000 spectrophotometer (Thermo Fisher Scientific, Waltham, MA, USA). The Hieff NGS Ultima Dual-mode mRNA Library Prep Kit (compatible with Illumina platform, Yeasen, China) was utilized to construct the sequencing library. The library fragments were purified using AMPure XP beads (Beckman Coulter, USA). Subsequently, the cDNA products were amplified by polymerase chain reaction (PCR), and finally sequenced on an Illumina HiSeq2500 genomic sequencer by Biomarker Biotechnology Co., Ltd (*Song et al., 2013*).

**Table 1  The experimental design of this study.**

| Group | Sample | Species | Experimental conditions |
|---|---|---|---|
| | CK1 | *Prunus persica* | Pre-processing comparison |
| CT0 | CK2 | *Prunus persica* | Pre-processing comparison |
| | CK3 | *Prunus persica* | Pre-processing comparison |
| | T1_1 | *Prunus persica* | Cutting-7d/control |
| CT1 | T1_2 | *Prunus persica* | Cutting-7d/control |
| | T1_3 | *Prunus persica* | Cutting-7d/control |
| | T2_1 | *Prunus persica* | Cutting-7d/substrate temperature-19 °C |
| CT2 | T2_2 | *Prunus persica* | Cutting-7d/substrate temperature-19 °C |
| | T2_3 | *Prunus persica* | Cutting-7d/substrate temperature-19 °C |
| | T3_1 | *Prunus persica* | Cutting-7d/substrate temperature-22 °C |
| CT3 | T3_2 | *Prunus persica* | Cutting-7d/substrate temperature-22 °C |
| | T3_3 | *Prunus persica* | Cutting-7d/substrate temperature-22 °C |
| | T4_1 | *Prunus persica* | Cutting-7d/substrate temperature-25 °C |
| CT4 | T4_2 | *Prunus persica* | Cutting-7d/substrate temperature-25 °C |
| | T4_3 | *Prunus persica* | Cutting-7d/substrate temperature-25 °C |
| | T5_1 | *Prunus persica* | Cutting-7d/substrate temperature-28 °C |
| CT5 | T5_2 | *Prunus persica* | Cutting-7d/substrate temperature-28 °C |
| | T5_3 | *Prunus persica* | Cutting-7d/substrate temperature-28 °C |
| | CK_14_1 | *Prunus persica* | Cutting-14d/control |
| CT6 | CK_14_2 | *Prunus persica* | Cutting-14d/control |
| | CK_14_3 | *Prunus persica* | Cutting-14d/control |
| | T6_1 | *Prunus persica* | Cutting-14d/substrate temperature-19 °C |
| CT7 | T6_2 | *Prunus persica* | Cutting-14d/substrate temperature-19 °C |
| | T6_3 | *Prunus persica* | Cutting-14d/substrate temperature-19 °C |
| | T7_1 | *Prunus persica* | Cutting-14d/substrate temperature-22 °C |
| CT8 | T7_2 | *Prunus persica* | Cutting-14d/substrate temperature-22 °C |
| | T7_3 | *Prunus persica* | Cutting-14d/substrate temperature-22 °C |
| | T8_1 | *Prunus persica* | Cutting-14d/substrate temperature-25 °C |
| CT9 | T8_2 | *Prunus persica* | Cutting-14d/substrate temperature-25 °C |
| | T8_3 | *Prunus persica* | Cutting-14d/substrate temperature-25 °C |
| | T9_1 | *Prunus persica* | Cutting-14d/substrate temperature-28 °C |
| CT10 | T9_2 | *Prunus persica* | Cutting-14d/substrate temperature-28 °C |
| | T9_3 | *Prunus persica* | Cutting-14d/substrate temperature-28 °C |
| | CK_21_1 | *Prunus persica* | Cutting-21d/control |
| CT11 | CK_21_2 | *Prunus persica* | Cutting-21d/control |
| | CK_21_3 | *Prunus persica* | Cutting-21d/control |
| | T10_1 | *Prunus persica* | Cutting-21d/substrate temperature-19 °C |
| CT12 | T10_2 | *Prunus persica* | Cutting-21d/substrate temperature-19 °C |
| | T10_3 | *Prunus persica* | Cutting-21d/substrate temperature-19 °C |
| | T11_1 | *Prunus persica* | Cutting-21d/substrate temperature-22 °C |

**Table 1** (*continued*)

| Group | Sample | Species | Experimental conditions |
|-------|--------|---------|--------------------------|
| CT13 | T11_2 | *Prunus persica* | Cutting-21d/substrate temperature-22 °C |
|  | T11_3 | *Prunus persica* | Cutting-21d/substrate temperature-22 °C |
|  | T12_1 | *Prunus persica* | Cutting-21d/substrate temperature-25 °C |
| CT14 | T12_2 | *Prunus persica* | Cutting-21d/substrate temperature-25 °C |
|  | T12_3 | *Prunus persica* | Cutting-21d/substrate temperature-25 °C |
|  | T13_1 | *Prunus persica* | Cutting-21d/substrate temperature-28 °C |
| CT15 | T13_2 | *Prunus persica* | Cutting-21d/substrate temperature-28 °C |
|  | T13_3 | *Prunus persica* | Cutting-21d/substrate temperature-28 °C |

## RNA-sequencing data analysis

Three biological replicates were set for each sample during RNA-sequencing. The statistical power of this experimental design, calculated in RNASeqPower is 0.84. By processing the raw data, we eliminated reads containing adapters, poly-N sequences, and low-quality reads to obtain high-quality clean reads. All subsequent downstream analyses were conducted based on these high-quality clean reads. Subsequently, we aligned these clean reads to the reference genome sequence (*Prunus persica*) and annotated them accordingly. To achieve rapid and accurate alignment of clean reads against the reference genome and to obtain their precise genomic coordinates, we utilized the HISAT2 v2.0.5 software (default parameters) (*Mortazavi et al., 2008*). The HISAT2 index was constructed from the reference genome Fasta file using default settings (hisat2-build). This configuration uses dynamic scoring for mismatch tolerance without explicit threshold specification. Furthermore, the SAMtools analysis tool is used to analyze the alignment status of specific regions (such as intergenic regions, exon regions, intron regions). Subsequently, transcript assembly was conducted using StringTie v1.3.6 (*Pertea et al., 2015*) in three stages: Per-sample assembly: stringtie <sample.bam>-p 4 -G <reference.gtf>-o <sample.gtf>; Merge assemblies across samples: stringtie –merge -G <reference.gtf>-l novel -o merged.gtf sample1.gtf sample2.gtf ...; Novel transcript identification: gffcompare -R -r <reference.gtf>-o gffcompare merged.gtf. Transcripts with no overlap to reference annotations (class code "u") were retained as novel genes. The quantitative analysis was conducted using the featureCounts v1.5.0-p3 tool within the subread software (*Liao, Smyth & Shi, 2014*). Set the minimum mapping quality score to 10. Finally, gene expression levels were estimated based on fragments per kilobase of transcript per million mapped reads (FPKM) (*Trapnell et al., 2010*). The DESeq2 R package (v 1.20.0) was adopted to conduct a significant analysis of the gene expression differences of the rootstocks of peach under different treatment conditions (*Anders & Huber, 2010*). *P*-values were adjusted for the false discovery rate (FDR) using the Benjamini–Hochberg method. The genes that were identified by DESeq2 with an adjusted *P*-value $\leq 0.05$ and $|\log_2 FC| \geq 1$ were regarded as DEGs. The Gene Ontology (GO) functional enrichment analysis and Kyoto Encyclopedia of Genes and Genomes (KEGG) pathway enrichment analysis of the DEGs sets were conducted using the clusterProfiler software (*Yu et al., 2012*).

## Weighted gene co-expression network construction

The gene co-expression network was constructed using the WGCNA R package. The Pearson correlation coefficient is computed based on gene expression profiles across diverse samples, followed by transformation into an adjacency matrix using a weighting function (*Wang et al., 2023*). The scale-free topology criterion was used to determine the soft threshold power (β), thereby ensuring that the gene expression matrix met the requirements of a scale-free network. A minimal module size of 30, a merge cut height of 0.25, and a area assign threshold of 0. Subsequently, visualization of gene modules is performed using Cytoscape.

# RESULTS

## Root phenotype analysis

We conducted a 40-day phenotypic analysis of the root systems of peach rootstock cuttings propagated under different substrate temperatures of 19 °C, 22 °C, 25 °C and 28 °C. The results revealed that root development was optimal under the condition of 25 °C (Fig. 1A). We further conducted statistics and analysis on the rooting rate of the rootstocks at 40 days post-cutting (Fig. 1B). The results indicated that the highest rooting rate, 91%, was obtained at a substrate temperature of 25 °C. Consequently, we can conclude that the optimal substrate temperature for the propagation of peach rootstocks by cutting is 25 °C.

## Overview of RNA-Seq data

We conducted an in-depth sequencing analysis on 45 libraries of peach rootstocks under three different cutting periods (7, 14, and 21 days) and five substrate temperature conditions (ambient temperature (CK), 19 °C, 22 °C, 25 °C, and 28 °C). This sequencing effort yielded over 2.1 billion raw reads, with an average of approximately 45 million reads per library (Table 2). After rigorous filtering process, RNA-Seq technology generated between 39 million and 48 million high-quality clean reads for each sample, with an average Q20 value exceeding 98.62% and a Q30 value exceeding 95.79% (Table 2). These high-quality data were utilized for all subsequent expression analyses. The filtered and cleaned reads were successfully aligned to the reference genome of *Prunus persica*, with an average genome alignment rate of 91.02% (Table 2). The multiple mapped category accounted for the largest proportion of 1.92% (CT0, CT1), while the unique mapped category had the lowest proportion of 82.1% (CT14). Furthermore, we conducted detailed statistical analysis on the alignment regions of each sample and found that exon regions accounted for up to 96.26% of the total number of reads (File S1).

We performed a comprehensive analysis of gene expression levels within our dataset using the featureCounts tool from the subread package. Specifically, File S2 presents a detailed read count expression matrix for 26,978 individual genes across various samples. Notably, the three biological replicates of the samples exhibited high Pearson correlation coefficients ($R^2 \geq 0.773$, Fig. 2A), indicating strong reproducibility in our data. To gain further insights into the differences and relationships between the samples, we employed principal component analysis (PCA) separated samples into 16 distinct clusters corresponding to the 15 experimental treatments (5 substrate temperatures ×3

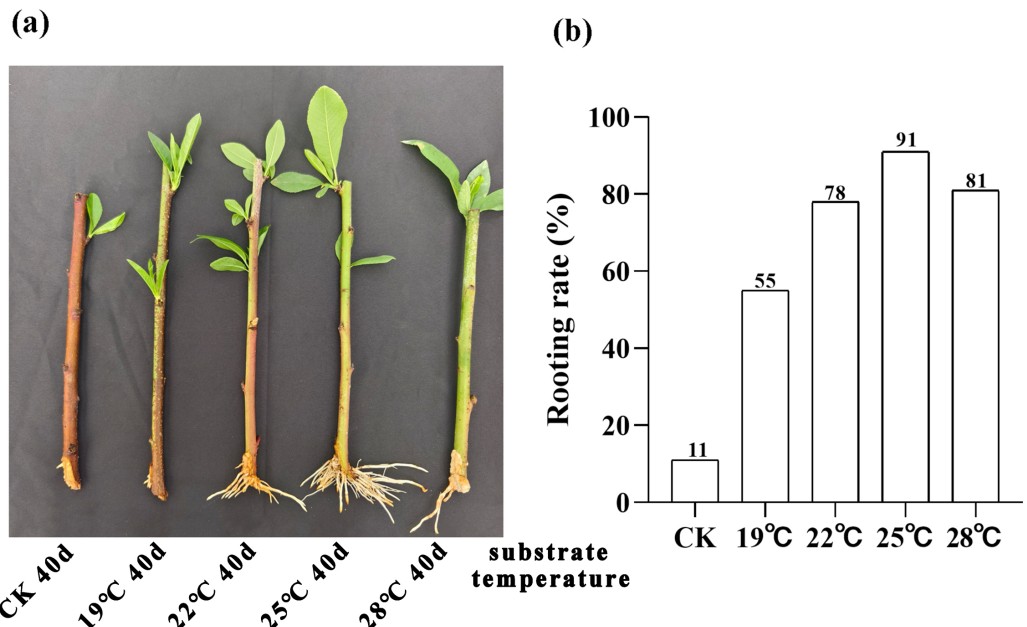

**Figure 1 Analysis of root phenotype in peach rootstock propagated by cutting under different substrate temperatures (19 °C, 22 °C, 25 °C, and 28 °C).** (A) Phenotypic analysis of root system in peach rootstock propagated by cuttings over a 40-day period at different substrate temperatures (19 °C, 22 °C, 25 °C, and 28 °C). (B) Statistics and analysis of rooting rate of peach rootstock cuttings after 40 days at different substrate temperatures. The vertical axis represents the rooting rate, while the horizontal axis represents different substrate temperatures (19 °C, 22 °C, 25 °C, and 28 °C).

cutting periods) plus the untreated control (CT0). Among them, peach rootstock cuttings incubated for 7 days at different substrate temperatures (ambient temperature (CK), 19 °C, 22 °C, 25 °C, 28 °C) are classified as CT1, CT2, CT3, CT4, and CT5, respectively; those incubated for 14 days at different substrate temperatures are classified as CT6, CT7, CT8, CT9, and CT10; while another set of cuttings incubated for 21 days are classified as CT11, CT12, CT13, CT14, and CT15. The control group is classified as CT0 (Fig. 2B and File S3). Furthermore, we employed various visualization techniques, including violin plots and boxplots to conduct a comprehensive and meticulous examination of the gene expression distribution across the samples (Figs. 2C–2D).

## Screening and analysis of DEGs

We conducted a detailed comparative analysis on 12 groups of DEGs ($P$-value $\leq 0.05$, $|\log_2 FC| \geq 1$). We compared CT1 with CT2, CT3, CT4, and CT5; CT6 with CT7, CT8, CT9, and CT10; CT11 with CT12, CT13, CT14, and CT15 respectively. Overall, the number of DEGs increased with the rise of substrate temperature at 7 days and 21 days of cutting. However, during the 14-day cutting treatment period, the increase in DEGs continued until 25 °C (Fig. 3A). Specifically, the highest count of DEGs in peach rootstock cuttings was observed on the 7th and 21st days after cutting at substrate temperatures of 25 °C and 28 °C. Conversely, the peak number on the 14th day post-cutting was noted at substrate
**Table 2 Summary of RNA sequencing data.**

| Group | Sample | Total raw reads | Total clean reads | Mapped to genome | Q20 (%) | Q30 (%) |
|---|---|---|---|---|---|---|
| | CK1 | 47,820,518 | 47,018,584 | 43,367,878 (92.24%) | 98.66 | 95.95 |
| CT0 | CK2 | 47,896,372 | 47,054,904 | 43,524,528 (92.5%) | 98.71 | 96.03 |
| | CK3 | 49,075,334 | 47,491,586 | 43,579,725 (91.76%) | 98.7 | 96.03 |
| | T1_1 | 46,112,880 | 45,301,060 | 41,701,330 (92.05%) | 98.87 | 96.52 |
| CT1 | T1_2 | 46,513,084 | 45,726,832 | 41,989,079 (91.83%) | 98.32 | 94.97 |
| | T1_3 | 48,004,348 | 46,895,460 | 40,816,357 (87.04%) | 98.65 | 96.02 |
| | T2_1 | 46,008,382 | 44,923,344 | 41,368,746 (92.09%) | 98.72 | 96.07 |
| CT2 | T2_2 | 41,222,534 | 40,417,280 | 37,188,481 (92.01%) | 98.29 | 94.89 |
| | T2_3 | 47,780,536 | 46,894,280 | 43,239,306 (92.21%) | 98.64 | 95.84 |
| | T3_1 | 49,306,966 | 48,448,074 | 44,320,546 (91.48%) | 98.75 | 96.15 |
| CT3 | T3_2 | 43,377,072 | 42,720,960 | 39,675,545 (92.87%) | 98.59 | 95.71 |
| | T3_3 | 47,114,448 | 46,115,510 | 42,722,806 (92.64%) | 98.73 | 96.07 |
| | T4_1 | 47,472,882 | 4,6615,394 | 42,620,563 (91.43%) | 98.59 | 95.7 |
| CT4 | T4_2 | 41419734 | 40721474 | 37,493,665 (92.07%) | 98.63 | 95.82 |
| | T4_3 | 46,931,744 | 46,132,170 | 42,280,235 (91.65%) | 98.58 | 95.7 |
| | T5_1 | 42,571,156 | 41,766,276 | 38,366,128 (91.86%) | 98.68 | 95.94 |
| CT5 | T5_2 | 40,391,866 | 39,734,218 | 36,152,071 (90.98%) | 98.51 | 95.45 |
| | T5_3 | 47,855,332 | 46,983,982 | 42,885,665 (91.28%) | 98.62 | 95.77 |
| | CK_14_1 | 45,846,246 | 44,855,450 | 40,900,606 (91.18%) | 98.64 | 95.84 |
| CT6 | CK_14_2 | 44,378,214 | 43,377,076 | 39,447,161 (90.94%) | 98.33 | 94.99 |
| | CK_14_3 | 49,966,392 | 48,015,634 | 43,531,783 (90.66%) | 98.84 | 96.46 |
| | T6_1 | 44,398,656 | 43,761,882 | 39,847,771 (91.06%) | 98.66 | 95.9 |
| CT7 | T6_2 | 47,909,356 | 46,768,074 | 42,655,348 (91.21%) | 98.67 | 95.93 |
| | T6_3 | 45,331,290 | 43,896,236 | 40,036,895 (91.21%) | 98.66 | 95.91 |
| | T7_1 | 40,947,492 | 40,247,198 | 36,715,915 (91.23%) | 98.6 | 95.75 |
| CT8 | T7_2 | 40,677,260 | 39,979,808 | 36,710,211 (91.82%) | 98.68 | 95.97 |
| | T7_3 | 40,433,808 | 39,794,474 | 36,550,697 (91.85%) | 98.57 | 95.6 |
| | T8_1 | 47,231,542 | 46,321,786 | 42,117,224 (90.92%) | 98.62 | 95.79 |
| CT9 | T8_2 | 41,345,866 | 40,678,696 | 37,076,433 (91.14%) | 98.62 | 95.79 |
| | T8_3 | 44547176 | 43796224 | 39,825,654(90.93%) | 98.63 | 95.83 |
| | T9_1 | 44,651,492 | 43,904,988 | 40,171,431 (91.5%) | 98.68 | 95.96 |
| CT10 | T9_2 | 43,849,526 | 43,155,586 | 39,536,447 (91.61%) | 98.66 | 95.9 |
| | T9_3 | 39,875,306 | 39,036,404 | 35,861,001 (91.87%) | 98.55 | 95.59 |
| | CK_21_1 | 46,027,828 | 45,074,462 | 40,851,282 (90.63%) | 98.64 | 95.85 |
| CT11 | CK_21_2 | 47,123,826 | 46,200,950 | 42,096,191 (91.12%) | 98.59 | 95.7 |
| | CK_21_3 | 40,978,358 | 39,970,156 | 36,344,705 (90.93%) | 98.31 | 94.95 |
| | T10_1 | 48,580,358 | 47,360,950 | 43,021,908 (90.84%) | 98.67 | 95.93 |
| CT12 | T10_2 | 47,158,522 | 46,329,548 | 42,013,549 (90.68%) | 98.61 | 95.8 |
| | T10_3 | 49,484,862 | 48,556,220 | 44,159,009 (90.94%) | 98.56 | 95.62 |
| | T11_1 | 48,290,834 | 47,349,840 | 43,256,047 (91.35%) | 98.67 | 95.93 |
| CT13 | T11_2 | 48,314,980 | 47,500,624 | 42,973,556 (90.47%) | 98.49 | 95.38 |
| | T11_3 | 46,036,442 | 45,205,652 | 41,274,112 (91.3%) | 98.65 | 95.87 |

**Table 2** (*continued*)

| Group | Sample | Total raw reads | Total clean reads | Mapped to genome | Q20 (%) | Q30 (%) |
|-------|--------|-----------------|-------------------|------------------|---------|---------|
| CT14 | T12_1 | 42,609,072 | 41,682,398 | 36,962,679 (88.68%) | 98.65 | 95.88 |
|      | T12_2 | 48,868,252 | 47,932,620 | 40,140,961 (83.74%) | 98.59 | 95.74 |
|      | T12_3 | 41,867,310 | 41,154,606 | 37,344,121 (90.74%) | 98.51 | 95.46 |
| CT15 | T13_1 | 47,484,014 | 46,618,360 | 41,306,853 (88.61%) | 98.71 | 96.03 |
|      | T13_2 | 48,005,162 | 46,647,020 | 41,825,243 (89.66%) | 98.66 | 95.9 |
|      | T13_3 | 46,150,870 | 44,995,606 | 40,631,894 (90.3%) | 98.7 | 96 |
| **Group** | **Sample** | 2,185,245,500 | 2,141,129,916 | | | |

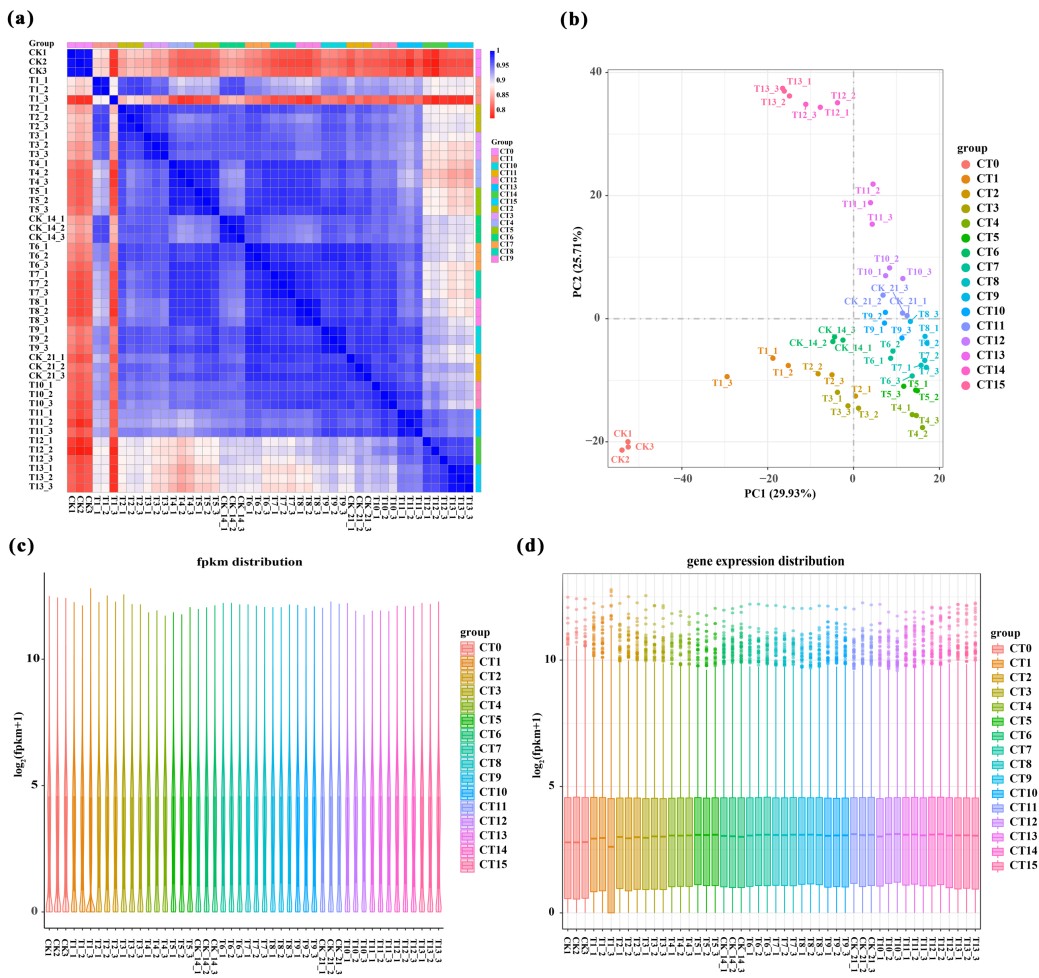

**Figure 2  Overview of RNA-Seq data.** (A) Pearson correlation analysis between samples. Values close to 1 indicate a strong positive correlation, values close to −1 indicate a strong negative correlation, and values close to 0 indicate no linear relationship. (B) 2D result plots of principal component analysis for various samples. Each point in the plot represents a sample, and its position reflects its scores along the first two principal components. (C–D) Violin plots and boxplots are used to visualize the distribution of gene expression levels within a sample. The same color was used for the three biological replicates of the same treatment, and different colors represented different treatments.

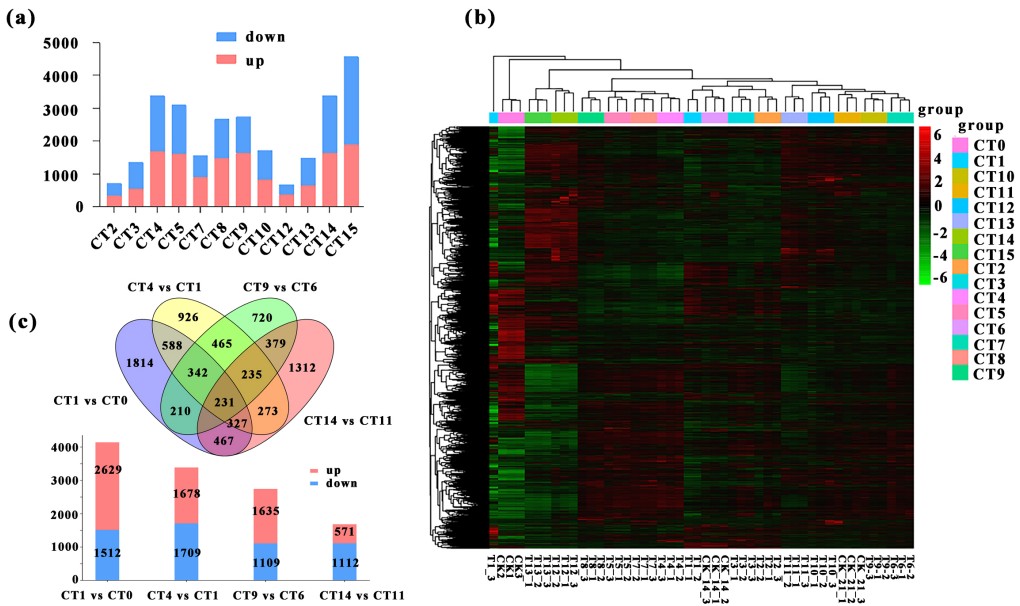

**Figure 3** **Comparative analysis of DEGs under different substrate temperatures treated by different cutting periods.** (A) The number of up and down-regulated DEGs between the 12 comparisons. CT2, CT3, CT4 and CT5 represent the up-regulated and down-regulated genes when the cuttings 7 d and the substrate temperature were 19 °C, 22 °C, 25 °C and 28 °C, respectively. CT 7, CT 8, CT 9 and CT 10 represent the up-regulated and down-regulated genes when the cuttings 14 d and the substrate temperature were 19 °C, 22 °C, 25 °C and 28 °C, respectively. CT 12, CT 13, CT 14 and CT 15 represent the up-regulated and down-regulated genes when the cuttings 21 d and the substrate temperature were 19 °C, 22 °C, 25 °C and 28 °C, respectively. (B) Heatmap analysis was conducted for 14,380 DEGs across various comparisons. The same color was used for the three biological replicates of the same treatment, and different colors represented different treatments. (C) Venn diagram of the total number of DEGs between the four comparisons (CT1 *vs* CT0, CT 4 *vs* CT 1, CT 9 *vs* CT 6, and CT 14 *vs* CT1). The bar chart represents the number of up-regulated and down-regulated genes in the four comparison groups. CT 0: control at 0 d ambient; CT1: control at 7 d ambient; CT4: 7 d at 25 °C; CT6: control at 14 d ambient; CT 9: 14 d at 25 °C; CT11: control at 21 d ambient; CT 14: 21 d at 25 °C.

temperatures ranging from 22 °C to 25 °C (Fig. 3A). From the significance analysis of upregulated and downregulated genes, the optimal substrate temperatures range for peach rootstock cuttings at 7- and 21-days post-insertion was determined to be 25 °C to 28 °C, whereas for cuttings at 14 days post-insertion, the optimal range shifted to 22 °C to 25 °C. Notably, the samples collected on the 21st day after cutting showed more prominent variations in differential gene expression levels under different substrate temperatures (Fig. 3A). Based on the statistical analysis of rooting rates for cuttings at various substrate temperatures (Fig. 1) and the statistical analysis of the number of DEGs, we conclude that the optimal substrate temperature for peach rootstock cuttings is 25 °C. Subsequently, we performed a heatmap analysis on the 14,380 DEGs identified in various comparisons (Fig. 3B).

More notably, there were 231 DEGs shared among the four comparisons, while 1,814, 926, 720, and 1,312 DEGs were uniquely expressed in the comparisons of CT1 *vs* CT0,

CT4 *vs* CT1, CT9 *vs* CT6, and CT14 *vs* CT11, respectively. Among them, the meanings represented by each treatment are: CT0: control at 0 d ambient; CT1: control at 7 d ambient; CT4: 7 d at 25 °C; CT6: control at 14 d ambient; CT9: 14 d at 25 °C; CT11: control at 21 d ambient; CT14: 21 d at 25 °C. Furthermore, in the comparison between CT1 and CT0, there were 2,629 upregulated genes and 1,512 downregulated genes. In the comparison between CT4 and CT1, there were 1,678 upregulated genes and 1,709 downregulated genes. Similarly, in the comparison between CT9 and CT6, there were 1,635 and 1,109 single genes were upregulated and downregulated, respectively. In the comparison between CT14 and CT11, there were 571 single genes were upregulated, with 1,112 single genes downregulated (Fig. 3C).

## GO enrichment and KEGG pathway analysis of DEGs

Based on the above conclusions, we will focus on conducting in-depth research on cuttings that have been propagated for 7, 14, and 21 days at a substrate temperature of 25 °C. These cuttings will be compared with the control group, and subjected to GO and KEGG pathway analysis. When comparing CT1 *vs* CT4, the most significantly enriched GO terms in biological processes were "microtubule-based movement", "cellular process", and "photosynthesis". Among the cellular components, three GO terms "thylakoid part", "photosynthetic membrane", and "photosystem" were highly enriched. In the comparisons of CT6 *vs* CT9, the most significantly enriched GO terms in biological processes were "response to stress" and "defense response". When comparing CT11 *vs* CT14, the most significantly enriched GO terms in biological processes were "movement of cell or subcellular component", "microtubule-based process", and "DNA-dependent DNA replication" (Fig. 4A). Among them, the meanings represented by each treatment are: CT1: control at 7 d ambient; CT4: 7 d at 25 °C; CT6: control at 14 d ambient; CT9: 14 d at 25 °C; CT11: control at 21 d ambient; CT14: 21 d at 25 °C. Across these comparisons, "copper ion binding" and "microtubule motor activity" were significantly enriched in terms of molecular function (Fig. 4A).

KEGG enrichment analysis was also conducted to identify the enriched metabolic pathways. The results indicated that significantly enriched KEGG pathways were mainly associated with photosynthesis processes, specifically including "Starch and sucrose metabolism" (Fig. 4B). Furthermore, several other enriched pathways related to hormones and metabolites were identified, including "Flavonoid biosynthesis", "Plant hormone signal transduction", and "Zeatin biosynthesis" (Fig. 4B).

## Identification DEGs involved in the formation of AR-related pathways

We systematically screened for DEGs that may be involved in AR formation under various cutting substrate temperatures and cutting periods. Previous studies have shown that *ARFs* regulate the development of adventitious roots by specifically binding to auxin response elements (*Tao et al., 2023*). In this study, we successfully identified five key *ARFs* (gene id: 18775811, 18774376, 18772269, 18791569, and 18786733). During the process of adventitious root formation, these genes increased by 1.33 to 3.46 times compared with the control group. Notably, the *LATERAL ORGAN BOUNDARIES DOMAIN (LBD)*

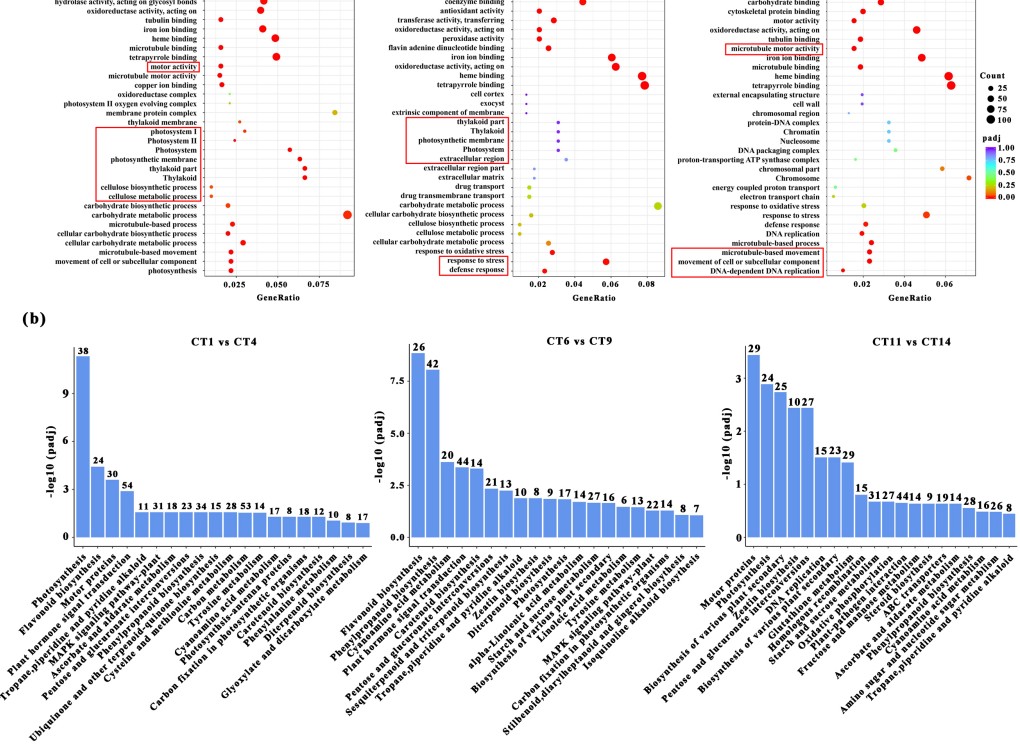

**Figure 4  GO enrichment and KEGG pathway analysis of DEGs.** (A) The GO enrichment analysis results between the four comparisons (CT1 *vs* CT4, CT6 *vs* CT9 and CT11 *vs* CT14). The red boxes represent the most significantly enriched biological processes between the two comparison groups. (B) The KEGG pathway analysis results between the four comparisons (CT1 *vs* CT4, CT6 *vs* CT9, and CT11 *vs* CT14). CT1: control at 7 d ambient; CT4: 7 d at 25 °C; CT6: control at 14 d ambient; CT9: 14 d at 25 °C; CT11: control at 21 d ambient; CT14: 21 d at 25 °C.

genes, as the primary downstream targets of ARFs, have been shown to be involved in the process of adventitious root formation (*Kirolinko et al., 2024*). In this screening, we found six differentially expressed *LBD* genes (gene id: 18774258, 18767664, 18777846, 18789146, 18786806, and 18774612) with varying expression patterns across different samples. Notably, the expression levels of *LBDs* in the cutting group were 1.36 to 6.85 times higher than those in the control group. The *SAUR* genes constitute the largest group of specific auxin-responsive factors participating in root development (*Zhou et al., 2024*). Among the screened 10 *SAURs* (gene id: 18770094, 18787544, 18793348, 18783824, 18789144, 18768105, 18783913, 18,784,823, 18792443, and 18785952), their expression levels exhibited significant differences compared to control samples, with the highest fold change being 42.09. On the other hand, members of the auxin-responsive *GH3* family play a crucial role in regulating auxin homeostasis through the synthesis of auxin conjugates in higher plants (*Caño et al., 2018*). Notably, the *GH3* gene (gene id: 18768891) showed prominent differential expression across various sample tests (Fig. 5). In summary, these
genes closely associated with auxin signaling may collectively participate in the formation of AR.

In *A. thaliana*, genes related to root formation are influenced by auxin, such as *LRP1*, *RGF*, and *AIR12* (*Singh et al., 2020*; *Shinohara, 2021*; *Gibson & Todd, 2015*). During the period of adventitious root formation, the expression level of the root-related gene *AUR3* (gene ID: 18786669) was significantly increased compared to the control group, with an increase of up to 3.8 times. Concurrently, during the AR formation period, the maximum expression levels of *LRP1* (gene ID: 18786210, and 18781220) increased by 18.2-fold and eight-fold, respectively. Additionally, four *RGF1* genes (gene ID: 18769572, 18790800, 18777415, and 18775914) exhibit increased expression during AR formation. Furthermore, the rooting-related gene *AIR9* (gene ID: 18780861) was also significantly upregulated during the formation of AR. Its expression reached the maximum value 7 days after cutting at 28 °C, increasing by 2.58 times compared to the control group (Fig. 5). These findings suggest that cellular division and proliferation activities are significantly enhanced during the development of ARs in peach cuttings.

Furthermore, the *NAM/ATAF1/2/CUC2* (NAC) transcription factor and the *APETALA2 (AP2)* genes are also widely involved in multiple aspects of plant organ development, such as root stem cell development and cell differentiation (*Li et al., 2019*; *Guyomarc'h, Boutté & Laplaze, 2021*). Through in-depth analysis, we have identified eight differentially expressed *NAC* genes (gene ID: 18784579, 18787178, 18768537, 18770366, 18790018, 18772621, 18789474, and 18790535) and six differentially expressed members of the *AP2* genes (gene ID: 18777768, 18773161, 9978767, 18773090, 18766891, and 18773658). During the process of adventitious root formation, the maximum fold changes of *NAC* and *AP2* genes compared with the control group were 4.61-fold and 7.79-fold, respectively (Fig. 5). The changes in the expression of these genes indicate that their regulatory roles in the formation of adventitious roots.

## Impact of substrate temperatures on adventitious root formation in peach hardwood cuttings *via* WGCNA

In order to precisely screen out the genes closely related to the response of cutting substrate temperature and period, we first evaluated the number of candidate genes. If the total number of genes did not exceed 45,000, we used the expression data of all genes as the basis for the subsequent weighted gene co-expression network analysis. After this screening procedure, we finally determined 26,978 genes, which became the core for constructing the weighted gene co-expression network. We selected the power value corresponding to $R^2$ of 0.8 as the soft threshold (File S4).

Based on the correlation of expression levels between genes, the construction of clustering trees, and the stability of modules, WGCNA analysis precisely divided all genes into 68 modules (Fig. 6A), with the number of genes in each module ranging from 34 to 4,072. Among these, 44 modules exhibited a high correlation with the samples mentioned in this study ($R > 0.90$, Fig. 6B). Notably, we found an extremely close correlation between the CK1 (CT0) sample and the MEmediumorchid module ($R = 0.994572$), and a tight correlation between the CK3 (CT0) sample and the MEdarkolivegreen module ($R = 0.996928$).

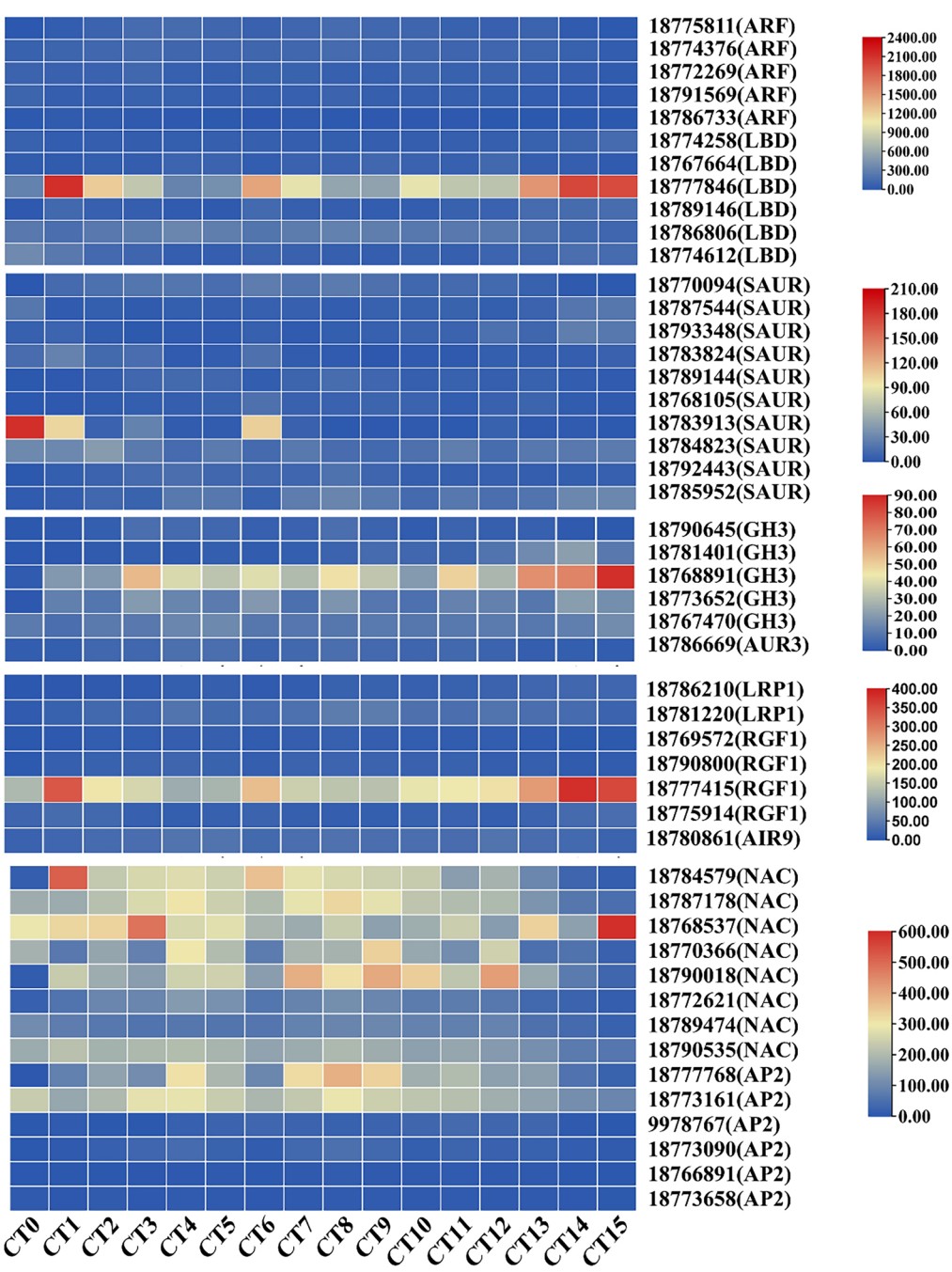

**Figure 5  Heatmap of gene expression related to root development and auxin signaling during the AR formation process in peach cuttings.** In this heatmap, each row represents a specific gene, while each column corresponds to a different sample or time point during the AR formation. The color intensity within each cell indicates the level of gene expression, with red colors signifying higher expression levels and blue colors representing lower expression levels.

Additionally, significant correlations were also observed between the T6-1 sample and the MEpaleturquoise module ($R = 0.994412$), as well as between the T9-2 sample and the MEplum module ($R = 0.994312$). Based on the clear heatmap of FPKM values, four modules exhibited distinct expression patterns at specific time points (Fig. 6C). Further analysis revealed that these four modules contained various transcription factors involved in the formation of AR-related genes, including four members of the *WRKY* family, eight members of the *ERF* family, six members of the *NAC* family, 11 members of the *bHLH* family, five members of the *bZIP* family, and 14 members of the *MYB* family (File S5). These transcription factors play crucial regulatory roles in the formation of AR, and their high expression levels during the corresponding periods fully support this point.

## DISCUSSION

The primary method of reproduction for woody plants is cuttage, which not only significantly enhances efficiency but also effectively preserves various valuable genetic traits (*Bannoud & Bellini, 2021*). In practical production operations, agronomic management measures such as cutting periods, substrate temperatures, and the quality of the cuttings themselves all have a notable influence on the formation process of root systems in cuttings (*Hilo et al., 2017*). In this study, comparative DEG analysis revealed that the optimal substrate temperature for peach rootstock cuttings (25 °C) aligns with maximal rooting rates observed after 40 days of cutting propagation. This finding resolves inconsistencies in prior studies regarding temperature optima for GF677 rootstocks (*Tsipouridis, Thomidis & Michailides, 2005*; *Eliwa & Wahba, 2018*). Further GO and KEGG analyses demonstrated that rooting performance differences under varying conditions are primarily driven by auxin pathway fluctuations. Specifically, our transcriptome data identified 26 auxin- and root development-related genes (*e.g.*, *ARFs*, *LBDs*, *SAURs*, *GH3*, *AUR3*, *LRP1*) and 22 transcription factors (*e.g.*, *WRKYs*, *ERFs*, *NACs*) that were overlooked in earlier physiological studies. These findings provide a molecular framework for adventitious root formation in peach rootstocks.

In the field of horticultural propagation techniques, the formation of adventitious roots is a crucial developmental stage. Auxin, serving as a universal regulatory factor in controlling root development and structural establishment, plays a crucial role in this process (*Overvoorde, Fukaki & Beeckman, 2010*). Studies have demonstrated that auxin can effectively induce adventitious root formation in various plants, such as *Camellia sinensis* (*Wei et al., 2019*), tomato (*Guan et al., 2019*), rice (*Lin & Sauter, 2019*), and apple (*Bai et al., 2020*). Notably, research focusing on Mango (*Mangifera indica L.*) has reported that overexpressing *MiARF2* inhibits the growth of roots and hypocotyls in *A. thaliana* (*Wu et al., 2011*). In our research, we screened five differentially expressed *ARFs*. During the process of adventitious root formation, these genes increased by 1.33 to 3.46 times compared with the control group. This phenomenon suggests that *ARFs* play different regulatory roles under varying cutting environmental conditions. However, further comprehensive research is required to elucidate the molecular mechanisms underlying these specific functions of ARFs. In *A. thaliana*, the activation of *LBD16* expression initiates organ development by

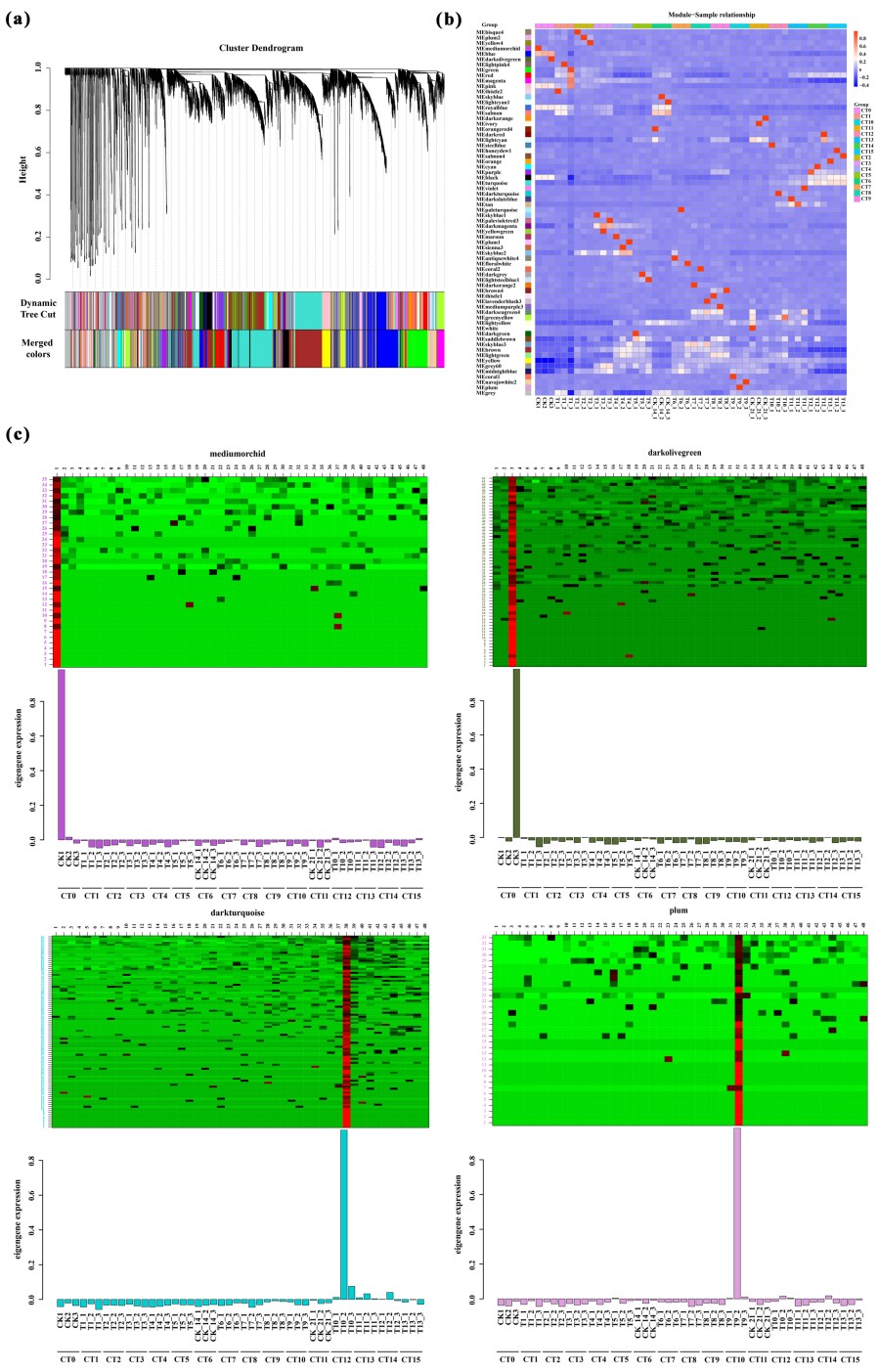

**Figure 6  Weighted gene co-expression network analysis.** (A) Clustering and module delineation. The *X*-axis represents different genes, and each branch represents a single gene. The *Y*-axis (Height) represents the similarity of the clusters. A lower height indicates (continued on next page...)

**Figure 6 (…continued)**
a higher degree of similarity between the genes. Each color represents a gene module. (B) Module-and-Sample correlation. This analysis can quantify the degree of association between the gene expression patterns within each module and specific sample groups. The horizontal axis represents the CT0-CT15 treatment, and the vertical axis represents the module. Red represents positive correlation, while blue represents negative correlation. (C) Heatmap of gene expression within the modules is used to visualize the relationships between gene expression patterns and sample groups. The horizontal axis represents the CT0-CT15 treatment. The vertical axis represents the expression of the eigengene in modules mediumorchid, darkolivegreen, darkturquoise and plum.

promoting cell proliferation and establishing root primordium identity (*Liu et al., 2018*). According to our RNA-seq analysis results, we identified six differentially expressed *LBD* genes, with their expression levels upregulated by approximately 1.36- to 6.85-fold in the cutting groups compared to the control group. According to reports, *SAUR36* controls adventitious root development in poplar *via* the auxin pathway (*Liu et al., 2018*). Based on transcriptome data, we screened for 10 *SAUR* genes, among which the highest expression was 42.09 times higher than that of the control samples. These results indicate that *ARF*, *SAUR*, and *LBD* may have conserved roles in the process of AR formation. Some GH3 proteins bind to auxin and negatively regulate root or AR development (*Gutierrez et al., 2012*). Among our results, the expression level of *GH3* (gene ID: 18768891) varied most significantly across all samples.

Through WGCNA analysis, we identified several transcription factors involved in the formation of AR-related genes, including *WRKYs*, *ERFs*, *NACs*, *bHLHs*, *bZIPs*, and *MYBs*, which may play a defensive role by regulating the expression of downstream genes. WRKY proteins exert dynamic roles in numerous plant processes involving responses to abiotic and biotic stresses (*Nuruzzaman et al., 2016*). The transcription factor *WRKY75* modulates hydrogen peroxide levels to regulate the development of adventitious roots, lateral buds, and callus tissues in poplar (*Zhang et al., 2022*). As an auxin-responsive transcription factor, *OsAP2/ERF* plays a pivotal role in promoting the growth and development of adventitious roots in various plant species such as rice (*Neogy et al., 2019*) and poplar (*Trupiano et al., 2013*). *NAC1* has been identified as a key regulator of lateral root development, with its expression mechanism closely related to auxin signaling pathways during lateral root growth (*Xie et al., 2000*). However, recent reports have found that *NAC1* plays a role in the regeneration of adventitious roots in leaf explants triggered by auxin (*Chen et al., 2016*). All the above evidence further confirms the concept that NAC participates in different regulatory pathways during root development. Furthermore, this study identified 8 differentially expressed *NACs* and 6 differentially expressed *AP2* members. *bHLHs* promote plant root development and play a crucial role in gibberellin metabolism and hormone regulation (*Du et al., 2023*; *Guo et al., 2023*). Evidence suggests that *bZIP* significantly contributes to auxin-regulated root growth by binding to downstream genes and regulating auxin-related transcriptional activity (*Zhang et al., 2020*). *MYBs* can regulate the growth and development of plant roots through jasmonic acid signaling pathways, ROS/PCD-dependent pathways, and abscisic acid response mechanisms (*Wang et al., 2024a*; *Wang et al., 2024b*; *Tong et al., 2024*; *Pan et al., 2024*). However, the molecular regulatory roles

of these transcription factors in the formation of adventitious roots in peach rootstocks remain to be further investigated.

## CONCLUSIONS

This study investigated the influence of cutting periods and substrate temperatures on adventitious root formation in peach rootstocks. Phenotypic analysis determined that the most suitable substrate temperature for peach rootstock cuttings is 25 °C. Transcriptome data revealed different gene sets regulated by cutting periods and substrate temperatures, thereby identifying a group of potential regulatory genes involved in adventitious root formation. These genes include auxin-related genes, root development-related genes, and some transcription factors. These findings offer new perspectives and clues for understanding the molecular mechanisms underlying adventitious root formation in peach rootstocks.

### Funding

This work was supported by the grants from the National Natural Science Foundation of China Project (No. 32360717) and the Modern Agricultural Industry Technology System Construction Special Project (No. CARS-30-1-6). The funders had no role in study design, data collection and analysis, decision to publish, or preparation of the manuscript.

### Grant Disclosures

The following grant information was disclosed by the authors:
The National Natural Science Foundation of China Project: No. 32360717.
The Modern Agricultural Industry Technology System Construction Special Project: No. CARS-30-1-6.

### Competing Interests

The authors declare there are no competing interests.

### Author Contributions

- Fan Zhang conceived and designed the experiments, authored or reviewed drafts of the article, and approved the final draft.
- Hong Wang conceived and designed the experiments, analyzed the data, prepared figures and/or tables, and approved the final draft.
- Chenbing Wang performed the experiments, analyzed the data, prepared figures and/or tables, and approved the final draft.
- Xiaoshan Wang analyzed the data, authored or reviewed drafts of the article, and approved the final draft.
- Jiaxuan Ren conceived and designed the experiments, prepared figures and/or tables, and approved the final draft.
- Meimiao Guo conceived and designed the experiments, authored or reviewed drafts of the article, and approved the final draft.
## Data Availability

The raw measurements are available in the Supplementary Files.

The RNA-Seq datasets are available at CNCB BioProject: PRJCA039237.

## Supplemental Information

Supplemental information for this article can be found online at http://dx.doi.org/10.7717/peerj.20015#supplemental-information.

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
