# Peer review of "Transcriptome analysis of substrate temperature effects on adventitious root formation in peach rootstocks"

_PeerJ, doi:10.7717/peerj.20015_

## Round 0.1 · original submission · Major Revisions

Add only relevant references suggested by the reviewer.

·

Basic reporting

Dear Authors, I reviewed the manuscript titled "A histology-based study of the effect of different substrate temperatures on adventitious root formation in peach hardwood cuttings". The manuscript demonstrated the differential gene expression patterns of GF677 rootstock cuttings treated with 200 mg•L-1 IBA under various substrate temperatures (unheated, 19 ℃, 22 ℃, 25 ℃, and 28 ℃) and cutting periods (7, 14, and 21 days).

I think that the author's opinion is very interesting, and the content is reasonable. However, this article raises some questions for me.

1- The research title is inconsistent with its content.

2- GF677 is a peach rootstock, not a peach.

3- Prunus rootstocks have a poor rooting capacity, making it challenging to propagate them on a large scale using hardwood cuttings. Do you think treating the GF677 hardwood cutting by dipping in 200 mgL-1 IBA for 20 s is sufficient to induce adventitious root formation?

Please refer to Eliwa, G.I., Sayed, S.N., Guirguis, N.S., Wahba, M.M. (2018). Comparative studies on the propagation of some imported peach rootstocks by using hardwood cuttings. HortScience J. Suez Canal Univ. 7 (2), 99–106. http://dx.doi.org/10.21608/hjsc.2018.59117
El-Boray, M. S.; Iraqi, M.A.; Samra, N.R. and Eliwa, G. I. (1995). Studies on rooting hardwood cuttings of Meit Ghamr peach cultivar. J. Agric. Sci. Mansoura Univ. 20 (12): 5127–5127. https://www.researchgate.net/publication/313724689
Eliwa, G. I. (1994). Physiological studies on peach trees. PhD. Thesis, Fac. Of Agric., Mansoura Univ. http://dx.doi.org/10.13140/RG.2.2.14215.65444

4- Lines 18 and 19: substrate temperatures (unheated, 19 #, 22 #, 25 #, and 28 #). Could you please explain what "unheated" means? Do you agree with me that the temperature differences are not precise and are not worth much?
Lines 90 and 92: What do you mean by “different cutting days”? Please clarify that, and when did you take the hardwood cuttings?

5- Lines 107 to 109: A cutter was used to create a 40 to 45° bevel at the base near the basal bud. The top of the cutting was cut flat, and it was advisable to apply a plant wound-healing agent to the flat surface. That is not true. The bottom of the cutting was cut flat close to the basal bud, and the top was beveled at a 40 to 45° angle.

6- Lines 121 to 128: RNA library construction and high-throughput sequencing
Please let me know when the samples are taken. Put the reference in there.

7- The only measurement made on the rooted cuttings was the rooting percentage, which is insufficient to examine the impact of the study's treatments, which are the basis for gene expression.

Experimental design

Not clear

Validity of the findings

The findings are not satisfactory

Additional comments

The only measurement made on the rooted cuttings was the rooting percentage, which is insufficient to examine the impact of the study's treatments, which are the basis for gene expression.

·

Basic reporting

Although the manuscript is generally well-structured, it needs more precise definitions and consistent terminology usage throughout. Although the headings of the sections (Abstract, Introduction, Methods, Results, Discussion) are appropriate, readers may find it challenging to follow each section because terms like "cutting periods," "substrate temperature," and control labels (such as "CK1," "CT5") are not always defined or applied consistently. Axis labels, legends, and data points must all be readable at high resolution (≥300 dpi), and figure and table captions should be more detailed. To prevent redundancy, reference citations should be condensed and acronyms (like IBA, AR, and WGCNA) should be explained when used for the first time.

Experimental design

Critical details are missing, but the overall design—treating GF677 cuttings with 200 mg·L⁻¹ IBA under five substrate temperatures and sampling at 7, 14, and 21 days—is sound. The controls' labeling scheme (e.g., "CK1," "CK7," "CK14," and "CK21") is inconsistent and needs to be clarified so that each CT code corresponds unambiguously to its temperature and time point. Furthermore, details regarding greenhouse conditions (ambient temperature, photoperiod, and humidity) are not provided, and it is unclear if RNA was extracted from individual cuttings or pooled basal segments. To guarantee complete reproducibility, software versions and important parameter settings (such as DESeq2 FDR correction criteria, HISAT2 mismatch thresholds, and WGCNA module-detection parameters) must be supplied.

Validity of the findings

Interpretation depends on consistent statistical and methodological rigor, but the core findings—that 25 °C yields the highest rooting rate and enriches auxin-related DEGs—are tenable. In addition to presenting summary statistics (such as the number of mapped reads per sample and the DEG counts at each temperature/time) in tables or figures rather than solely in text, DEG thresholds should be set as FDR-adjusted p < 0.05 with |log₂FC| ≥ 1. Justification is required for the comparison of particular CT pairs (e.g., CT1 vs. CT0, CT2 vs. CT6), and overlap in DEGs should distinguish between up- and down-regulation. Clearer explanations of module-trait correlations (such as precise R and p-values) and a succinct tabulation of important transcription factors by module would improve WGCNA results. The authors can increase trust in their purported temperature-dependent regulatory networks by addressing these issues.

Additional comments

ABSTRACT:

Line 17: At the first mention, spell out "indole-3-butyric acid (IBA)."

Line 19: The abstract mentions a rooting rate "at 25 °C for 40 days," but it also reports RNA-seq sampling at 7, 14, and 21 days. Please include a brief statement stating that the final rooting percentage was recorded at 40 days, but transcript profiling took place at earlier times.

Line 25: "26 plant hormone signaling pathways (ARFs, LBDs, SAURs, GH3)" and "22 AR formation–related pathways (AUR3, LRP1, RGF1, AIR9, AP2, NAC)" are mentioned in the abstract. These are gene families rather than entire pathways. Please update to refer to them as "genes" or "gene families" to align with the methodologies.

INTRODUCTION:
Lines 64–87: Although the review of auxin-responsive families is comprehensive, it might be too detailed for an introduction.

Line 64: If auxin "serves as an effective inducer," has already been mentioned in the first paragraph; don't reiterate it.

Standardize gene-family formatting according to journal style (e.g., use all caps for families, italics for gene names).

Condense the goals and novelty into a single, concise statement at the conclusion.

MATERIALS AND METHODS:
The experimental procedures are generally well‐organized, but several critical details (sample labeling, pooling strategy, greenhouse conditions, software versions, and analysis parameters) are missing or ambiguous.

The text states that samples at 7 d, 14 d, and 21 d were labeled “T1–T5,” “CK14/T6–T9,” and “CK21/T10–T13,” respectively, and then notes “CK1 represents the untreated control group.” However, it's unclear how the 7d control was handled, and neither "CK1" nor "CK7" was previously defined.
Although it is stated in the Methods that "each replicate contained 30 randomly selected healthy cuttings," it is unclear if RNA was taken from individual cuttings or pooled tissue. Likewise, there is no description of the precise tissue area (e.g., basal 2 cm containing AR initials) that was utilized for RNA extraction.

Although the heating-wire setup and substrate composition (perlite:peat moss:vermiculite = 1:1:1) are described, the general greenhouse conditions (ambient temperature range, photoperiod, light intensity, and relative humidity) are not.

Although the pipeline (HISAT2-SAMtools-StringTie-featureCounts-DESeq2) is suitable, the Methods section is missing important parameter settings and software version numbers (e.g., DESeq2 normalization method, HISAT2 index version, mismatch allowance, minimum mapping quality, StringTie assembly options).

Although the effect-size assumptions (e.g., expected fold change, dispersion, FDR) are not given, the reported RNASeqPower value of 0.84 is helpful.

It's unclear if raw p-values or Benjamini-Hochberg-adjusted p-values are meant by the "P-value < 0.05" cutoff.

RESULTS:
A thorough multi-layered analysis connecting temperature, rooting rate, transcriptomics, DEG identification, and WGCNA is presented in the results. However, reconciling sample/library counts, consistently defining CT codes, summarizing numerical data in tables/figures instead of lengthy text, and explicitly stating statistical thresholds (e.g., adjusted p-values) would all significantly improve clarity.
According to the text, cuttings were evaluated at 20 days (Fig. 1a) and then at 40 days (Fig. 1 b) to determine the "rooting rate." Please explain why the 20-day data were deemed "unsatisfactory" and provide evidence for evaluating the final rooting at 40 days, when the Methods only mentioned sampling at 7, 14, and 21 days.

Three time points × five temperatures are reported for "39 libraries" in the paper. But 45 libraries, plus controls, are equal to three replicates × three times × five temperatures. Kindly explain the number of controls included or reconcile this disparity (for example, "39 treatment libraries + 6 control libraries = 45 total").

CT1–CT15 are grouped into 16 clusters in the description, but 16 clusters for 39 samples seems excessive (16 clusters imply many small groups). Make it clear if "16 clusters" means different sample-group centroids or just that the samples were separated by temperature and time (e.g., five CT groups at 7 d, five at 14 d, five at 21 d, plus one control cluster).

There is no obvious key to remind readers which CT number correlates to which temperature and time in comparisons like "CT1 vs. CT0," "CT2 vs. CT6," etc. Restate each new section in brief (e.g., “CT1: 7 d at 19 °C; CT6: 14 d at 19 °C; CTo: control at 7 d ambient”).
"49 DEGs were shared among the four comparisons, while 3,065, 443, 682, and 135 DEGs were uniquely expressed in CT1 vs CT0, CT2 vs CT6, CT2 vs CT11, and CT2 vs CT1, respectively," the text reads. Indicate if these figures pertain to total DEGs, downregulated DEGs, or upregulated DEGs for clarity. Additionally, briefly explain why these four specific pairwise comparisons (CT1 vs. CT0, CT2 vs. CT6, etc.) were chosen. Do they represent baseline vs. same-time temperature shift, or something else?

DISCUSSION:
Lines 313–322 restate most of the points in the Results. Instead of restating the entire background ("woody plant reproduction is via cuttage…"), the focus of the discussion should be on interpreting your new findings. To illustrate how your comparative DEG analysis surpasses previous research, for instance, remove or simplify the generalizations about "cutting, substrate temperature, cutting duration" (lines 314–320).
"Rooting rates recorded at 40 days post-cutting" to "rooting rates after 40 days of cutting propagation" (line 319).

Line 323 uses the phrase "differential gene expression under different cutting environmental conditions" multiple times; instead, consider "differential expression under varying temperature and time conditions."
The statement "WRKY proteins regulate stress-responsive pathways" in line 352 could be shortened to "WRKY proteins exert dynamic roles in numerous plant processes involving responses to abiotic and biotic stresses."

FIGURES:
Please provide high-resolution versions (300 dpi or higher) of the current figures so that all details, including axis labels, legends, and fine structures, are clear and legible. The figures are low resolution and appear pixelated, making labels and data points difficult to read.

---

## Round 0.2 · accepted · Accept

The authors have addressed all the comments, and the reviewer is satisfied with the revised version; hence, the manuscript can be accepted for publication.

·

Basic reporting

"A histology-based study of the effect of different substrate temperatures on adventitious root formation in peach hardwood cuttings." The manuscript demonstrated the differential gene expression patterns of GF677 rootstock cuttings treated with 200 mg•L⁻¹ IBA under various substrate temperatures (unheated, 19℃, 22℃, 25℃, and 28℃) and cutting periods (7, 14, and 21 days).
Overall, the article became clear; the findings were illustrated by the adequate background and introduction. The authors revised the paper in response to the reviewers' comments, and I believe the manuscript is now ready for publication.

Experimental design

Enough information is provided to describe the methods. The design of the experiment is appropriate and correct.

Validity of the findings

All supporting information has been supplied; it is reliable, statistically sound, and under control.

Additional comments

The authors revised the paper in response to the reviewers' comments, and I believe the manuscript is now ready for publication.